# Parental Perceived Discrimination and Youth Participation in Out-Of-School Sports

**DOI:** 10.3390/children9121808

**Published:** 2022-11-24

**Authors:** Ryan D. Burns, You Fu

**Affiliations:** 1Department of Health and Kinesiology, University of Utah, Salt Lake City, UT 84112, USA; 2School of Public Health, University of Nevada, Reno, NV 89557, USA

**Keywords:** adolescents, children, physical activity, sexual identity, stigma

## Abstract

The aim of this study was to determine the associations between perceived parental discrimination and youth participation in out-of-school sports. We analyzed a representative sample of US children and adolescents from the 2020 National Survey of Children’s Health (*N* = 30,656; 6–17 years old; 49.0% female). The dependent variable was a (No/Yes) response item asking parents about their child’s participation in out-of-school sports. Two binary response items asked parents if they perceived that their child was ever treated or judged unfairly because of their race/ethnicity and because of their sexual orientation/gender identity. Weighted multiple logistic regressions examined correlations between the discrimination variables and out-of-school sports controlling for demographics covariates including family income. After covariate adjustment, perceived parental discrimination because of race/ethnicity associated with 1.57 times higher odds of child participation in sports (Adjusted Odds Ratio (AOR) = 1.57, 95%CI: 1.17–2.03, *p* = 0.002). Conversely, discrimination because of sexual orientation/gender identity correlated with 57% lower odds of child participation in sports (AOR (Adjusted Odds Ratio) = 0.43, 95%CI: 0.27–0.69, *p* < 0.001). After family income and other covariates were considered, perceived discrimination by parents because of their child’s race/ethnicity associated with higher probability of sports participation. Perceived discrimination by parents because of their child’s sexual orientation/gender identity associated with a lower probability of participating in sports.

## 1. Introduction

Daily structure and routine have been positively linked to better health behaviors, lower substance abuse, and better mental health [1,2,3,4]. Many structured activities can facilitate prevention of diseases related to lifestyle, which inherently require a lifetime of prevention strategies and treatment [5]. Structured and organized activities also provide opportunities for socialization that can enhance sense of connectiveness and enjoyment of a respective activity that can further motivate an individual for future participation [6,7,8]. Structured activities that involve physical activity, such as sports, can help prevent manifestation of excess sedentary behavior-related disease risk factors and can provide participants a sense of belonging, purpose, and achievement [9,10].

Within the young people, the Structured Days Hypothesis provides a framework for explaining how health behaviors fluctuate according to the structure of days [11]. Tenets of the Structured Days Hypothesis include that when youth are within a structured day with structured activities they are more likely to have better health behaviors compared to when they are not within a structured day [11,12]. For example, when youth are in school settings, they tend to accumulate higher levels of physical activity and consume more healthy meals compared to days with less structure [11,12]. A way to provide additional structure during out-of-school hours for youth is for participation in out-of-school sports.

Youth of a racial and ethnic minority, a sexual minority, or a trans or non-binary gender identity tend to have poorer health behaviors [13,14,15]. Minority youths also face frequent discrimination and stigma can preclude access to activities that provide daily structure, like out-of-school sports [16,17]. Barriers to participation in structured activity such as sports can include lack of awareness, exclusion, limited availability, and fear of social judgement, amongst others [18]. Previous work has shown that in adults, racial/ethnic and sexual and gender minority groups tend to participate in less structured activities during the workweek and on the weekends [19,20]. In youth, previous work has shown a lower prevalence of both physical activity and sports participation, especially among male and female racial/ethnic minority youth and among male sexual minority youth [21,22]. Minority youth also tend to participate less in organized activities and show a lower prevalence of being a volunteer compared to their non-minority counterparts [23,24]. However, it is unclear how perceived discrimination from a child’s parent associates with participation in out-of-school sports. Perceived discrimination by parents in childhood and adolescence may be a genesis for the activities minority youth participate in as they track into young adulthood. 

Parental support for physical activity and sport is needed for healthy child development and success within the sport environment [25,26]. However, perceived parental discrimination may increase psychological distress and thus lower support for child participation in various out-of-school activities. Previous research has shown that perceived discrimination associates with poorer health behaviors, but these effects are moderated by coping styles and emotional social support [16,27]. Perceived discrimination has also associated with lower perceived social support within some minority groups [28]. Parental perceived discrimination may lower child participation in out-of-school sports to avoid additional discrimination or unfairness from sports participation and to mitigate further psychological distress; however, different sources of discrimination may have differential associations with sports participation. Thus, the disparities in out-of-school sports participation among minority youth may partially manifest from perceived discrimination by parents. However, there are no studies linking perceived discrimination by parents to their child’s participation in out-of-school sports. Therefore, the aim of this study was to determine the correlation between parental perceived discrimination because of racial/ethnic minority status and sexual orientation/gender identity with child participation in out-of-school sports. It was hypothesized that parents who perceived discrimination because of their child’s race/ethnicity and sexual orientation/gender identity will associate with a lower probability of out-of-school sports. 

## 2. Materials and Methods

### 2.1. Participants

We analyzed a representative sample of youth whose parents completed the 2020 US National Survey of Children’s Health (NSCH; *N* = 30,656). Descriptive demographic characteristics of the analyzed sample (young people aged 6–17 years old) are communicated within Table 1. Approximately 50.9% of the participants were 12–17 years old and 49.0% of the sample were female. Most of the sample were White (66.5%) and of a Non-Hispanic or Latino ethnicity (73.7%). Most of the sample had a family poverty ratio above 200% (60.3%) and approximately 33.1% had overweight/obesity.

### 2.2. Sampling Procedures

Secondary analysis of the publicly available 2020 NSCH dataset was utilized to address the research question. The 2020 NSCH was provided online and via mail by the US Census Bureau. A random sample of selected addresses of US households were mailed instructions to access the online survey. Administrative data helped determine addresses with young people aged 0–17 years old. The households who were non-responsive to the online version were mailed a paper screening questionnaire [29,30]. The study used publicly available and deidentified data, did not qualify as human subjects research, and did not require institutional review board review.

### 2.3. National Survey of Children’s Health

The 2020 NSCH began on 27 July 2020 and continued until 22 January 2021. Online and paper respondents completed an initial screener questionnaire followed by completion of one of three versions of a topical questionnaire based on a randomly selected household child’s age: 0–5 years old, 6–11 years old, and 12–17 years old. Paper respondents received the topical questionnaire through the mail. A total of 42,777 topical questionnaires were completed for young people aged 0–17 years old (42% response rate); however, only young people aged 6–17 years old were used for the current study’s analysis (*N* = 30,656; 49.0% female). Child- and household-level sampling weights were used. Each weight was the product of a base sampling weight for each household, nonresponse adjustment, and raking adjustment. Weighted estimates were representative of the US child population [29,30]. 

The 2020 NSCH consisted of items asking parents about their child’s health, experience with healthcare providers, learning and schooling activities, after-school activities, the familial and household environment, amongst other topics. Seven new items were included on the 2020 version of the NSCH. One of these new items asked parents about discrimination (being treated or judged unfairly) because of their child’s sexual orientation or gender identity. Informed written consent was obtained from the participants [29,30].

### 2.4. Data Processing

The dependent variable was a recoded (No = 0, Yes = 1) response item asking parents about their child’s participation in out-of-school sports. The header was “During the past 12 months, did this child participate in:”. The specific item then asked, “A sports team or did they take sports lessons after school or on weekends?”. The discrimination covariates were two recoded (No = 0, Yes = 1) response items that asked parents about their perception of their child being treated or judged unfairly because of their race or ethnic group and because of their sexual orientation or gender identity. Headers for each of the two items included “To the best of your knowledge, has this child ever experienced any of the following?”. Specific items then asked, “Treated or judged unfairly because of their race or ethnic group?” and “Treated or judged unfairly because of their sexual orientation or gender identity?”. Additional covariates were included in the multivariable analyses to account for potential confounding associations. We included age, sex, race, ethnicity, family income, and child weight status based on calculated BMI %tile from parent-reported anthropometric data. 

### 2.5. Statistical Analysis

The descriptive statistics were presented as counts (*n*) and weighted percent’s (*%*). The complex multiple-stage NSCH sampling design was adjusted for within all analyses using Stata’s “svy:”prefix command. Taylor linearized variance estimations were calculated from a derived stratum variable crossing state of residence with a household identifier. The primary analysis consisted of weighted logistic regression models to determine the unadjusted and adjusted correlations between the discrimination covariates and sports participation. Weighted simple logistic regression models examined the unadjusted associations between each discrimination covariate by itself and out-of-school sports participation. Weighted multiple logistic regressions tested the correlations between the discrimination covariates together and out-of-school sports participation controlling for covariates. Two-way interactions between discrimination covariates were included to test for joint associations but were dropped if no significant associations were found. Missing data were imputed using multiple imputation methods pooled across 10 imputation models. We reported the unadjusted and adjusted odds ratios with 95% Confidence Intervals. Post estimation predicted probabilities were calculated from the weighted multivariable logistic regression model and displayed in Figure format. Alpha level was set a *p* < 0.05 and all analyses were conducted using Stata version 17.0 statistical software package (Statacorp., College Station, TX, USA).

## 3. Results

### 3.1. Descriptive Statistics

Descriptive statistics for the out-of-school sports participation variable and discrimination variables are communicated in Table 2. Most of the youth participated in out-of-school sports over the past year (53.0%). Approximately 6.9% of the sample were perceived as being treated unfairly because of their race/ethnicity and 1.4% because of their sexual orientation/gender identity.

### 3.2. Weighted Logistic Regression Models

Results from the weighted simple logistic regressions are communicated in Table 3. Unfair treatment or judgment because of sexual orientation/gender identity associated with lower odds for a child participating in out-of-school sports (OR = 0.41, 95%CI: 0.28–0.60, *p* < 0.001). Results from the weighted multivariable logistic regressions are reported in Table 4. After adjustment for demographic covariates, unfair treatment or judgment because of race/ethnicity associated with higher odds of child participation in out-of-school sports (AOR = 1.54, 95%CI: 1.17–2.03, *p* = 0.002). After adjustment for demographic covariates, unfair treatment or judgment because of sexual orientation/gender identity associated with lower odds of child sports participation (AOR = 0.43, 95%CI: 0.27–0.69, *p* < 0.001). No there no two-way interactions were statistically significant. Predicted probabilities of participation in out-of-school sports was 59% without perceived parental discrimination and 63% with perceived parental discrimination because of race/ethnicity. Conversely, predicted probabilities of participation in out-of-school sports was 60% without perceived parental discrimination and 35% with perceived parental discrimination because of sexual orientation/gender identity. These predicted probabilities are displayed in Figure 1.

## 4. Discussion

The study aim was to determine the association between perceived discrimination by parents because of their child’s race/ethnicity and sexual orientation/gender identity with child participation in out-of-school sports. The results indicated that there was a significant correlation between perceived discrimination by parents because of their child’s race or ethnicity and a higher likelihood of child participation in out-of-school sports. Conversely, perceived discrimination by parents because of their child’s sexual orientation or gender identity associated with a lower likelihood of their child participating in out-of-school sports. 

A salient result that supported our original hypothesis was that perceived discrimination by parents because of their child’s sexual orientation or gender identity associated with a lower likelihood of the child participating in out-of-school school sports. Sexual stigma has been observed within team sports settings and it thought to contribute to participation avoidance in individuals of a sexual minority or of a trans or non-binary gender identity [31,32,33]. This phenomenon does seem to be of higher prevalence in males of a sexual minority compared to females [21,22,33]. The low prevalence of sexual minority and/or trans or non-gender sports participation could originate through perceived stigma from parents. Parents themselves may perceive stigma or victimization because of their child’s sexual minority status or gender identity [34], and may have feelings of judgement, shame, and embarrassment if the child was placed in competitive sports environment with mostly heterosexual and cis-gender peers [35]. Feelings of discrimination and stigma by parents may mitigate support for their child to participate in out-of-school sports. It is unclear of these feelings of discrimination are shared between the parent and child and how different sources of perceptions of discrimination and stigma associate with both individual and team sports participation. 

Previous research has shown that adolescents of a sexual and/or gender minority also tend to have less interest in sports participation [32]. Lower interest or enjoyment in sports may be because of stigma or a feeling of a lack of connectiveness with heterosexual and cis-gender peers [36]. A lack of connection may lead to less enjoyment, intention, and actual participation in sports [37]; therefore, the tendency for sexual minority and trans-gender individuals to avoid participating in out-of-school sports may be because of the psychosocial factors related with Self-Determination Theory that drive behavior intention and behavior execution [38]. The lack of sports participation in these minority youths may lower the number of opportunities to make connection with peers and provides one less avenue to be physically active for a lifetime. Finding ways to reduce stigma and feelings of perceived discrimination is needed to facilitate enjoyment, interest, and participation in sporting activities in these marginalized groups of young people. 

Findings that opposed our original hypotheses were that parents who perceived discrimination because of their child’s race or ethnicity associated with a higher likelihood of that child participating in out-of-school sports. Previous research has shown that racial/ethnic disparities in health and health behaviors are because of several factors ranging from cultural barriers, language barriers, and unequal access to facilities [13]. Aligning with Fundamental Cause Theory, some have argued that the explanations for racial/ethnic disparities are largely due to socioeconomic status [39]. This is because socioeconomic status dictates access to the resources needed to participate in health behaviors, such as the organized and structured physical activity provided by out-of-school sports. This helps explain the findings of the current study as crude analyses had showed no association between perceived parental discrimination because of race/ethnicity and out-of-school sports participation. However, when family income and other covariates were considered, perceived parental discrimination because of race/ethnicity associated with higher odds out-of-school sports participation. When predicted probabilities were compared, there was a 4% difference in the probability of participation between groups (59% vs. 63%). Although these differences were relatively small compared to the differences in predicted probabilities due to perceived discrimination because of sexual orientation/gender identity, the findings support Fundamental Cause Theory when also considering that the family poverty ratio covariate had a strong correlation with out-of-school sports participation. Despite this, location of residence was not considered within the analysis. In a study by Williams et al. [40], it was found that youth residing in high poverty areas had worse health outcomes regardless of family income. Therefore, engagement in health behaviors via participation out-of-school sports may be more of a function of place of residence (low poverty vs. high poverty) than family income or socioeconomic status per se. This phenomenon should be examined in future research.

A higher likelihood of child out-of-school sports participation despite perceived parental discrimination is also supported by previous research, independent of family income or socioeconomic status. Snape et al. [41], suggested that many individuals of a racial/ethnic minority actively participated in sports despite the presence of perceived discrimination. Some racial/ethnic minority individuals have even used sports to proactively denounce racism and inspire others to participate in sports regardless of the presence of discrimination [42]. Hartman et al. [43] reported that some parents were interested in encouraging their children to participate in sports not only because of the potential health benefits it provides but also because sports is an opportunity for socialization and exposing their children to different racial/ethnic groups and cultures. For parents of a racial/ethnic minority, sport can be a unique opportunity for their children to be involved within the community, a potential benefit that can offset the negative effects of perceived racial/ethnic discrimination. Lastly, sports can provide structure within out-of-school settings and can prevent child anti-social behavior, violence, and drug use after school hours [44,45,46]. Parents may recognize these positive attributes of sports, despite the potential for discrimination and its associated psychological distress. 

There are alternative explanations for the observed findings. One potential alternative explanation is reverse causation. Youth who participate in sports may be more likely to experience discrimination and unfair treatment compared to youth to do not participate in out-of-school sports. Although there are several benefits of sports, participation does manifest potential for unfairness on several levels. Quality of coaching may mitigate this possibility, but it is unclear what the true directions of the observed associations are given the cross-sectional research design. Additionally, although we controlled for many covariates in our models, some covariates were not included that may have confounded the observed associations include household size (including the number of siblings), socioeconomic status of the schools, child perceived discrimination, and parental support for sport participation. Given the current analyses, it is unclear whether perceived discrimination by the parents leads to less support for child participation in sports. Parental support for sport and/or physical activity was not collected on the NSCH. These more complex associations should be explored in future research with moderation and mediation analyses.

There are practical implications of this study. One is that different sources of parental perceived discrimination seem to have different associations with child participation in out-of-school sports. Educating parents on the benefits of sports participation is needed to maintain support in the face of perceived discrimination. Socialization, engagement with the community, physical health benefits, in addition to reductions in anti-social behavior are all potential benefits of youth participation in sports. Additionally, both parental and child education on how to cope with discrimination and education on how parents or guardians can provide emotional support in the presence of discrimination may maintain sports participation in youth and potentially attenuate drop-out. Quality training of sport head and assistant coaches may help lower the probability of discrimination. Finally, cultivating an accepting and non-judgmental environment may especially be important for youth of a sexual minority or of a transgender identity in order to reduce stigma and to improve connectedness within the sport environment. 

The study’s limitations included the analysis of a cross-sectional dataset that inhibits making cause-and-effect inferences. Additionally, the directionality of the associations cannot be ascertained. The items related to both discrimination and participation in out-of-school were general but characteristic of population-level surveillance methods; therefore, specific information is not available pertaining to the degree and frequency of perceived discrimination and type and frequency of specific activities. All variables were collected using self-report; therefore, there is potential for response bias. Finally, although the results do generalize to the US child population, they do not generalize to the US adult (parent) population or to populations outside of the US. 

## 5. Conclusions

The results of this study provide evidence that perceived discrimination by parents because of their child’s sexual orientation or gender identity associates with a lower likelihood of their child participating in out-of-school sports. This could be explained by sexual stigma that is related to having a certain sexual orientation or having a specific gender identity and/or a lack of parental support or general child interest for sports participation in these minority groups. The associations that were observed between discrimination because of a youth’s race and ethnicity and a higher likelihood of participation in sports can be explained by the consideration of family income within the analysis and parental desire for their child to be socialized and to be a part of their community, despite the presence of discrimination. It is recommended that future research test the true directionality of the associations in addition to exploring how the intensity and frequency of perceived discrimination by both the parent and the child together associates with out-of-school sports participation. This study provides initial evidence that likelihood of child out-of-school sports participation may be different depending on the source of perceived parental discrimination as perceived discrimination because of race/ethnicity associated with a higher likelihood and because sexual orientation/gender identity associated with a lower likelihood of participation within a representative sample of young people.

## Figures and Tables

**Figure 1 children-09-01808-f001:**
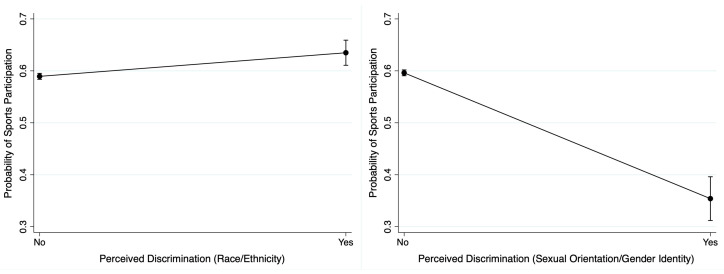
Adjusted predicted probabilities for out-of-school sports participation as a function of perceived parental discrimination because of the child’s race/ethnicity (**left**) and because of the child’s sexual orientation/gender identity (**right**). Note: Predicted probabilities adjusted for covariates.

**Table 1 children-09-01808-t001:** Characteristics of the analyzed sample of participants from the National Survey of Children’s Health.

Variable	Level	*n*	Weighted %s
Age (years)	6–11	13,097	49.1%
	12–17	17,539	50.9%
Sex	Male	15,853	51.0%
	Female	14,783	49.0%
Race	White	23,450	66.5%
	Black or African American	2384	14.3%
	American Indian/Alaskan Native	320	2.2%
	Asian	1766	4.8%
	Hawaiian/Pacific Islander	231	3.1%
	Two or More Races	2485	9.1%
Ethnicity	Non-Hispanic/Latino	26,519	73.7%
	Hispanic/Latino	4117	26.3%
Family Income (FPR)	0–99%	3798	17.9%
	100–199%	5131	21.8%
	200–399%	9353	29.6%
	400% or Greater	12,354	30.7%
Overweight/Obese	No	15,299	66.9%
	Yes	6212	33.1%

Note: FPR stands for Family Poverty Ratio.

**Table 2 children-09-01808-t002:** Descriptive statistics of the analyzed variables.

Variable	Level	*n*	Weighted %s
Sports Participation	No	12,368	47.0%
	Yes	17,774	53.0%
Unfair Treatment Race/Ethnicity	No	27,799	93.1%
	Yes	1622	6.9%
Unfair Treatment Sexual Orientation/Gender Identity	No	28,838	98.6%
	Yes	510	1.4%

**Table 3 children-09-01808-t003:** Unadjusted odds ratios from weighted simple logistic regressions.

Covariate	Level	OR	95% CI
Unfair Treatment Race/Ethnicity	No	Ref.	Ref.
	Yes	1.07	0.86–1.33
Unfair Treatment Sexual Orientation/Gender Identity	No	Ref.	Ref.
	Yes	**0.41 ^***^**	0.28–0.60

Note: OR stands for odds ratio; 95% CI stands for 95% Confidence Interval; estimates not adjusted for demographic covariates; bold denotes statistical significance, ^***^
*p* < 0.001.

**Table 4 children-09-01808-t004:** Adjusted odds ratios from the weighted multivariable logistic regression model.

Covariate	Level	AOR	95% CI
Unfair Treatment Race/Ethnicity	No	Ref.	Ref.
	Yes	**1.54 ^**^**	1.17–2.03
Unfair Treatment Sexual Orientation/Gender Identity	No	Ref.	Ref.
	Yes	**0.43 ^***^**	0.27–0.69
Age (years)	6–11	Ref.	Ref.
	12–17	0.94	0.80–1.10
Sex	Male	Ref.	Ref.
	Female	**0.70 ^***^**	0.62–0.81
Race	White	Ref.	Ref.
	Black or African American	0.92	0.75–1.13
	American Indian/Alaskan Native	**0.53 ^*^**	0.28–0.98
	Asian	**0.66 ^***^**	0.52–0.83
	Hawaiian/Pacific Islander	1.19	0.51–2.78
	Two or More Races	0.79	0.58–1.07
Ethnicity	Non-Hispanic/Latino	Ref.	Ref.
	Hispanic/Latino	**0.68 ^***^**	0.56–0.82
Family Income (FPR)	0–99%	Ref.	Ref.
	100–199%	1.08	0.85–1.38
	200–399%	**1.70 ^***^**	1.36–2.12
	400% or Greater	**3.14 ^***^**	2.54–3.83
Weight Status	Healthy Weight	Ref.	Ref.
	Overweight/Obese	**0.62 ^***^**	0.54–0.72

Note: AOR stands for adjusted odds ratio; 95% CI stands for 95% Confidence Interval; FPR stands for Family Poverty Ratio; estimates adjusted for all demographic covariates; bold denotes statistical significance, ^*^
*p* < 0.05, ^**^
*p* < 0.01, ^***^
*p* < 0.001.

## Data Availability

Data used in this study are publicly available and can be found at https://www.census.gov/programs-surveys/nsch/data/datasets.html (accessed on 1 March 2022).

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
