# Peer review of "Parental Perceived Discrimination and Youth Participation in Out-Of-School Sports"

_children, 2022, doi:10.3390/children9121808_

Round 1

Reviewer 1 Report

The article develops a theme of undeniable topicality and pertinence. However, by focusing the study on the effect of parental perceived discrimination, some ideas remain unexplained... The introduction does not objectively substantiate the importance of this topic. why is it important to study parental perceived discrimination? In the authors' opinion, or from existing studies, does this factor help or hinder the participation in sporting activities?
After description of the results it is verified that there is a positive relation in an aspect and negative in another... but the justification is merely speculative... There was not a question to know if the parents encourage the participation or not?
The article would benefit from some adjustments.

Author Response

The introduction does not objectively substantiate the importance of this topic. why is it important to study parental perceived discrimination? In the authors' opinion, or from existing studies, does this factor help or hinder the participation in sporting activities?

-Thank you. This important background information has now been added within the Introduction section (lines 105–123).

After description of the results, it is verified that there is a positive relation in an aspect and negative in another... but the justification is merely speculative... There was not a question to know if the parents encourage the participation or not?

-Thank you. The confounding factors that were not included in the analysis are now commented on within the Discussion section (lines 567–591).

Reviewer 2 Report

Thank you very much for sending me this manuscript. This is an important article and I believe this should be published. I have a few minor comments. 

1. The association between sport participation and unfair racial/ethnic treatment. I'd like to point to the possibility that there is a selection bias, making those who participate into sport activities are more likely to experience unfair treatment. However, as you did not find this trend with unfair treatment due to sexual minority status, I'd like to see the discussion on this particular effect.

2. The bivariate result for racial/ethnic treatment was not significant although this becomes significant in multivariate analysis. How do you explain this potential mediation? Please discuss potential mechanisms where this particular relationship may be influenced by other covariates.

3. How about policy implications? Considering that results are very important, situating these results within the existing policy context may carry significant policy contributions. Do you have any recommendations based on your results?

Author Response

The association between sport participation and unfair racial/ethnic treatment. I'd like to point to the possibility that there is a selection bias, making those who participate into sport activities are more likely to experience unfair treatment. However, as you did not find this trend with unfair treatment due to sexual minority status, I'd like to see the discussion on this particular effect.                                                        -Thank you. This is now commented on within the Discussion section (lines 567–591).

The bivariate result for racial/ethnic treatment was not significant although this becomes significant in multivariate analysis. How do you explain this potential mediation? Please discuss potential mechanisms where this particular relationship may be influenced by other covariates. -Thank you. This is now commented on within the Discussion section (lines 563–591).

How about policy implications? Considering that results are very important, situating these results within the existing policy context may carry significant policy contributions. Do you have any recommendations based on your results? -Thank you. A practical implications paragraph has now been added within the Discussion (lines 592–604).